# SYNERGISTIC INFORMATION RETRIEVAL: INTERPLAY BETWEEN SEARCH AND LARGE LANGUAGE MODELS

## ABSTRACT

Information retrieval (IR) plays a crucial role in locating relevant resources from vast amounts of data, and its applications have evolved from traditional knowledge bases to modern retrieval models (RMs). The emergence of large language models (LLMs) has further revolutionized the IR field by enabling users to interact with search systems in natural languages. In this paper, we explore the advantages and disadvantages of LLMs and RMs, highlighting their respective strengths in understanding user-issued queries and retrieving up-to-date information. To leverage the benefits of both paradigms while circumventing their limitations, we propose **InteR**, a novel framework that facilitates information refinement through synergy between RMs and LLMs. InteR allows RMs to expand knowledge in queries using LLM-generated knowledge collections and enables LLMs to enhance prompt formulation using retrieved documents. This iterative refinement process augments the inputs of RMs and LLMs, leading to more accurate retrieval. Experiments on large-scale retrieval benchmarks involving web search and low-resource retrieval tasks demonstrate that InteR achieves overall superior zero-shot retrieval performance compared to state-of-the-art methods, even those using relevance judgment.

## 1    INTRODUCTION

Information retrieval (IR) is an indispensable technique for locating relevant resources in a vast sea of data given ad-hoc queries (Mogotsi, 2010). It is a core component in knowledge-intensive tasks such as question answering (Karpukhin et al., 2020), entity linking (Gillick et al., 2019) and fact verification (Thorne et al., 2018). Over the years, the techniques of information retrieval have evolved significantly: from the traditional knowledge base (KB) (Lan et al., 2021; Gaur et al., 2022) to modern search engines (SEs) based on neural representation learning (Karpukhin et al., 2020; Yates et al., 2021), information retrieval has become increasingly important in our digital world. More recently, the emergence of cutting-edge large language models (LLMs; e.g., ChatGPT (OpenAI, 2022), GPT-4 (OpenAI, 2023), Bard (Google, 2023), LLaMA (Touvron et al., 2023a;b)) has further revolutionized the NLP community and given intriguing insights into IR applications as users can now interact with search systems in natural languages.

Over the decades, search engines like Google or Bing have become a staple for people looking to retrieve information on a variety of topics, allowing users to quickly sift through millions of documents to find the information they need by providing keywords or a query. Spurred by advancements in scale, LLMs have now exhibited the ability to undertake a variety of NLP tasks in a zero-shot scenario (Qin et al., 2023) by following instructions (Ouyang et al., 2022; Sanh et al., 2022; Min et al., 2022; Wei et al., 2022). Therefore, they could serve as an alternative option for people to obtain information directly by posing a question or query in natural languages (OpenAI, 2022), instead of relying on specific keywords. For example, suppose a student is looking to write a research paper on *the history of jazz music*. They could type in keywords such as *"history of jazz"* or *"jazz pioneers"* to retrieve relevant articles and sources. However, with LLMs, this student could pose a question like *"Who were the key pioneers of jazz music, and how did they influence the genre?"* The LLMs could then generate a summary of the relevant information and sources, potentially saving time and effort in sifting through search results.

As with most things in life, there are two sides to every coin. Both IR technologies come with their own unique set of advantages and disadvantages. LLMs excel in understanding the context and meaning behind user-issued textual queries (Mao et al., 2023), allowing for more precise retrieval of information, while RMs expect well-designed precise keywords to deliver relevant results. Moreover, LLMs have the capacity to directly generate specific answers to questions (Ouyang et al., 2022), rather than merely present a list of relevant documents, setting them apart from RMs. However, it is important to note that RMs still have significant advantages over LLMs. For instance, RMs can index a vast number of up-to-date documents (Nakano et al., 2021), whereas LLMs can only generate information that falls within the time-scope of the data they were trained on, potentially leading to hallucinated results (Shuster et al., 2021; Ji et al., 2023; Zhang et al., 2023a;b). Additionally, RMs can conduct quick and efficient searches through a vast amount of information on the internet, making them an ideal choice for finding a wide range of data. Ultimately, both paradigms have their own unique set of irreplaceable advantages, making them useful in their respective areas of application.

To enhance IR by leveraging the benefits of RMs and LLMs while circumventing their limitations, we consider bridging these two domains. Fortunately, we observe that textual information refinement can be performed between two counterparts and boost each other. On the one hand, RMs can gather potential documents with valuable information, serving as demonstrations for LLMs. On the other hand, LLMs generate concise summaries using well-crafted prompts, expanding the initial query and improving search accuracy. To this end, we introduce **InteR**, a novel framework that facilitates information refinement through synergy between RMs and LLMs. Precisely, the RM part of InteR receives the knowledge collection from the LLM part to refine and expand the information in the query. While the LLM part involves the retrieved documents from the RM part as demonstrations to enrich the information in prompt formulation. This two-step refinement procedure can be seamlessly repeated to augment the inputs of RM and LLM. Implicitly, we assume that the outputs of both components supplement each other, leading to more accurate retrieval.

We evaluate InteR on public large-scale retrieval benchmarks involving web search and low-resource retrieval tasks following prior work (Gao et al., 2023). The experimental results show that InteR can conduct zero-shot retrieval with overall better performance than state-of-the-art methods, even those using relevance judgment[1], and achieves new state-of-the-art zero-shot retrieval performance.

Overall, our main contributions can be summarized as follows:

- We introduce InteR, a novel IR framework bridging two cutting-edge IR products, search systems and large language models, while enjoying their strengths and circumventing their limitations.

- We propose iterative information refinement via synergy between retrieval models and large language models, resulting in improved retrieval quality.

- Evaluation results on zero-shot retrieval demonstrate that InteR can overall conduct more accurate retrieval than state-of-the-art approaches and even outperform baselines that leverage relevance judgment for supervised learning.

## 2 RELATED WORK

**Dense Retrieval**    Document retrieval has been an important component for several knowledge-intensive tasks (Voorhees et al., 1999; Karpukhin et al., 2020). Traditional techniques such as TF-IDF and BM25 depend on term matching and create sparse vectors (Robertson, 2009; Yang et al., 2017; Chen et al., 2017) to ensure efficient retrieval. After the emergence of pre-trained language models (Devlin et al., 2019; Liu et al., 2019), dense retrieval which encodes both queries and documents into low-dimension vectors and then calculates their relevance scores (Lee et al., 2019; Karpukhin et al., 2020), has recently undergone substantial research. Relevant studies include improving training approach (Karpukhin et al., 2020; Xiong et al., 2021; Qu et al., 2021), distillation (Lin et al., 2021; Hofstätter et al., 2021) and task-specific pre-training (Izacard et al., 2022; Gao & Callan, 2021; Lu et al., 2021; Gao & Callan, 2022; Xiao et al., 2022) of dense retrieval models which significantly outperform sparse approaches.

---

[1]In IR tasks, the relevance judgment illustrates the label of relevance between each pair of query and document, which is mainly used for supervised learning of an IR model.

**Zero-shot Dense Retrieval**   Many prior works consider training dense retrieval models on high-resource passage retrieval datasets like Natural Questions (NQ) (Kwiatkowski et al., 2019) (133k training examples) or MS-MARCO (Bajaj et al., 2016) (533k training examples) and then evaluating on queries from new tasks. These systems (Wang et al., 2022; Yu et al., 2022) are utilized in a transfer learning configuration (Thakur et al., 2021). However, on the one hand, it is time-consuming and expensive to collect such a vast training corpus. On the other hand, even MS-MARCO has limitations on commercial use and cannot be used in a wide range of real-world applications. To this end, recent work (Gao et al., 2023) proposes building zero-shot dense retrieval systems that require no relevance supervision (i.e., relevance label between a pair of query and document), which is considered "unsupervised" as the only supervision resides in the LLM where learning to follow instructions is conducted in earlier times (Sachan et al., 2022). In this work, we follow this zero-shot unsupervised setting and conduct information refinement through synergy between RMs and LLMs without any relevance supervision to handle the aforementioned issues.

**Enhance Retrieval Through LMs**   Recent works have investigated using auto-regressive language models to generate intermediate targets for better retrieval (Cao et al., 2021; Bevilacqua et al., 2022) while identifier strings still need to be created. Other works consider "retrieving" the knowledge stored in the parameters of pre-trained language models by directly generating text (Petroni et al., 2019; Roberts et al., 2020). Some researchers (Mao et al., 2021; Anantha et al., 2021; Wang et al., 2023) utilize LM to expand the query and incorporate these pseudo-queries for enhanced retrieval while others choose to expand the document (Nogueira et al., 2019). Besides, LMs can also be exploited to provide references for retrieval targets. For instance, GENREAD (Yu et al., 2023) directly generates contextual documents for given questions.

**Enhance LMs Through Retrieval**   On the contrary, retrieval-enhanced LMs have also received significant attention. Some approaches enhance the accuracy of predicting the distribution of the next word during training (Borgeaud et al., 2022) or inference (Khandelwal et al., 2020) through retrieving the k-most similar training contexts. Alternative methods utilize retrieved documents to provide supplementary context in generation tasks (Joshi et al., 2020; Guu et al., 2020; Lewis et al., 2020). WebGPT (Nakano et al., 2021) further adopts imitation learning and uses human feedback in a text-based web-browsing environment to enhance the LMs. LLM-Augmentor (Peng et al., 2023) improves large language models with external knowledge and automated feedback. REPLUG (Shi et al., 2023) prepends retrieved documents to the input for the frozen LM and treats the LM as a black box. Demonstrate–Search–Predict (DSP) (Khattab et al., 2022) obtains performance gains by relying on passing natural language texts in sophisticated pipelines between a language model and a retrieval model, which is most closely related to our approach. However, they rely on composing two parts with in-context learning and target on multi-hop question answering. While we aim at conducting information refinement via multiple interactions between RMs and LLMs for large-scale retrieval.

## 3   PRELIMINARY

**Document Retrieval: the RM Part**   Zero-shot document retrieval is a crucial component of search systems. Given the user query $q$ and the document set $D = \{d_1, ..., d_n\}$ where $n$ is the number of document candidates, the goal of a retrieval model (RM) is to retrieve documents that are relevant to satisfy the user's real search intent of the current query $q$. To accomplish such document retrieval, prior works can be categorized into two groups: sparse retrieval and dense retrieval. Both lines of research elaborate on devising the similarity function $\phi(q, d)$ for each query-document pair.

The sparse retrieval, e.g., TF-IDF and BM25, depends on lexicon overlap between query $q$ and document $d$. This line of RMs (Zhou et al., 2022; Thakur et al., 2021) ranks documents $D$ based on their relevance to a given query $q$ by integrating term frequency and inverse document frequency. Another works (Qu et al., 2021; Ni et al., 2022; Karpukhin et al., 2020) focus on dense retrieval that uses two encoding modules to map an input query $q$ and a document $d$ into a pair of vectors $\langle \mathbf{v_q}, \mathbf{v_d} \rangle$, whose inner product is leveraged as a similarity function $\phi$:

$$\phi(q, d) = \langle E_Q(q), E_D(d) \rangle = \langle \mathbf{v_q}, \mathbf{v_d} \rangle \tag{1}$$

Then the top-$k$ documents, denoted as $\bar{D}$ that have the highest similarity scores when compared with the query $q$, are retrieved efficiently by RMs regardless of whether the retrieval is sparse or dense.

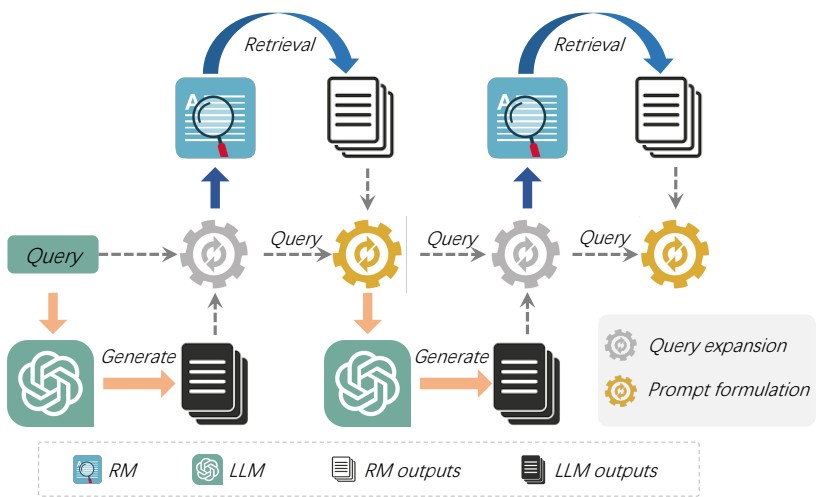

Figure 1: Overall architecture of InteR.

Noting that as for dense retrieval, following existing methods (Gao et al., 2023), we pre-compute each document's vector $\mathbf{v_d}$ for efficient retrieval and build the FAISS index (Johnson et al., 2019) over these vectors, and use Contriever (Izacard et al., 2022) as the backbone of query encoder $E_Q$ and document encoder $E_D$.

**Generative Retrieval: the LLM Part** Generative search is a new paradigm of IR that employs neural generative models as search indices (Tay et al., 2022; Bevilacqua et al., 2022; Lee et al., 2022). Recent studies propose that LLMs further trained to follow instructions could zero-shot generalize to diverse unseen instructions (Ouyang et al., 2022; Sanh et al., 2022; Min et al., 2022; Wei et al., 2022). Therefore, we prepare textual prompts $p$ that include instructions for the desired behavior to $q$ and obtain a refined query $q'$. Then the LLMs $G$ such as ChatGPT (OpenAI, 2022) take in $q'$ and generate related knowledge passage $s$. This process can be illustrated as follows:

$$s = G(q') = G(q \oplus p) \tag{2}$$

where $\oplus$ is the prompt formulation operation for $q$ and $p$. For each $q'$, if we sample $h$ examples via LLM $G$, we will obtain a knowledge collection $S = \{s_1, s_2, ..., s_h\}$.

## 4 INTER

On top of the preliminaries, we introduce **InteR**, a novel IR framework that iteratively performs information refinement through synergy between RMs and LLMs. The overview is shown in Figure 1. During each iteration, the RM part and LLM part refine their information in the query through interplay with knowledge collection (via LLMs) or retrieved documents (via RMs) from previous iteration. Specifically, in RM part, InteR refines the information stored in query $q$ with knowledge collection $S$ generated by LLM for better document retrieval. While in LLM part, InteR refines the information in original query $q$ with retrieved document $\bar{D}$ from RM for better invoking LLM to generate most relevant knowledge. This two-step procedure can be repeated multiple times in an iterative refinement style.

### 4.1 RM STEP: REFINING INFORMATION IN RM VIA LLM

When people use search systems, the natural way is to first type in a search query $q$ whose genre can be a question, a keyword, or a combination of both. The RMs in search systems then process the search query $q$ and retrieve several documents $\bar{D}$ based on their relevance $\phi(q, d)$ to the search query $q$. Ideally, $\bar{D}$ contains the necessary information related to the user-issued query $q$. However, it may include irrelevant information to query as the candidate documents for retrieval are chunked and

fixed (Yu et al., 2023). Moreover, it may also miss some required knowledge since the query is often fairly condensed and short (e.g., *"best sushi in San Francisco"*).

To this end, we additionally involve the generated knowledge collection $S$ from LLM in previous iteration and enrich the information included in $q$ with $S$. Specifically, we consider expanding the query $q$ by concatenating each $s_i \in S$ multiple times[2] to $q$ and obtaining the similarity of document $d$ with:

$$\begin{aligned}\phi(q, d; S) &= \phi([q; s_1; q; s_2; ...; q; s_h], d) \\ &= \langle E_Q([q; s_1; q; s_2; ...; q; s_h]), E_D(d) \rangle\end{aligned} \quad (3)$$

where $[\cdot; \cdot]$ is a concatenating operation for query expansion. Now the query is knowledge-intensive equipping with $S$ from LLM part that may be supplement to $q$. We hope the knowledge collection $S$ can provide directly relevant information to the input query $q$ and help the RMs focus on the domain or topic in user query $q$.

## 4.2 LLM Step: Refining Information in LLM via RM

As aforementioned, we can invoke LLMs to conditionally generate knowledge collection $S$ by preparing a prompt $p$ that adapts the LLM to a specific function (Eq. 2). Despite the remarkable text generation capability, they are also prone to hallucination and still struggle to represent the complete long tail of knowledge contained within their training corpus (Shi et al., 2023). To mitigate the aforementioned issues, we argue that $\bar{D}$, the documents retrieved by RMs, may provide rich information about the original query $q$ and can potentially help the LLMs make a better prediction.

Specifically, we include the knowledge in $\bar{D}$ into $p$ by designing a new prompt as:

```
Give a question {query} and its possible answering passages {passages}
Please write a correct answering passage:
```

where "{query}" and "{passages}" are the placeholders for $q$ and $\bar{D}$ respectively from last RM step:

$$s = G(q') = G(q \oplus p \oplus \bar{D}) \quad (4)$$

Now the query $q'$ is refined and contains more plentiful information about $q$ through retrieved documents $\bar{D}$ as demonstrations. Here we simply concatenate $\bar{D}$ for placeholder "{passages}", which contains $k$ retrieved documents from RM part for input of LLM $G$.

## 4.3 Iterative Interplay Between RM and LLM

In this section, we explain how iterative refinement can be used to improve both RM and LLM parts. This iterative procedure can be interpreted as exploiting the current query $q$ and previous-generated knowledge collection $S$ to retrieve another document set $\bar{D}$ with RM part for the subsequent stage of LLM step. Then, the LLM part leverages the retrieved documents $\bar{D}$ from previous stage of RM and synthesizes the knowledge collection $S$ for next RM step. A critical point is that we take LLM as the starting point and use only $q$ and let $\bar{D}$ be empty as the initial RM input. Therefore, the prompt of first LLM step is formulated as:

```
Please write a passage to answer the question.
Question: {query}
Passage:
```

We propose using an iterative IR pipeline, with each iteration consisting of the four steps listed below:

1. Invoke LLM to conditionally generate knowledge collection $S$ with prompt $q'$ on Eq. 4. The retrieved document set $\bar{D}$ is derived from previous RM step and set as empty in the beginning.

---

[2] In our preliminary study, we observed that concatenating each $s_i \in S$ multiple times to $q$ can lead to improved performance, as the query is the most crucial component in IR.

2. Construct the updated input for RM with knowledge collection $S$ and query $q$ to compute the similarity of each document $d$.

3. Invoke RM to retrieve the top-$k$ most "relevant" documents as $\bar{D}$ on Eq. 3.

4. Formulate a new prompt $q'$ by combining the retrieved document set $\bar{D}$ with query $q$.

The iterative nature of this multi-step process enables the refinement of information through the synergy between the RMs and the LLMs, which can be executed repeatedly $M$ times to further enhance the quality of results.

## 5 EXPERIMENTS

### 5.1 DATASETS AND METRICS

Following (Gao et al., 2023), we adopt widely-used web search query sets TREC Deep Learning 2019 (DL'19) (Craswell et al., 2020) and TREC Deep Learning 2020 (DL'20) (Craswell et al., 2021) which are based on the MS-MARCO (Bajaj et al., 2016). Besides, we also use six diverse low-resource retrieval datasets from the BEIR benchmark (Thakur et al., 2021) consistent with (Gao et al., 2023) including SciFact (fact-checking), ArguAna (argument retrieval), TREC-COVID (bio-medical IR), FiQA (financial question-answering), DBPedia (entity retrieval), and TREC-NEWS (news retrieval). It is worth pointing out that we do not employ any training query-document pairs, as we conduct retrieval in a zero-shot setting and directly evaluate our proposed method on these test sets. Consistent with prior works, we report MAP, nDCG@10, and Recall@1000 (R@1k) for TREC DL'19 and DL'20 data, and nDCG@10 is employed for all datasets in the BEIR benchmark.

### 5.2 BASELINES

**Methods without relevance judgment** We consider several zero-shot retrieval models as our main baselines, because we do not involve any query-document relevance scores (denoted as *w/o relevance judgment*) in our setting. Particularly, we choose heuristic-based lexical retriever BM25 (Robertson & Zaragoza, 2009), BERT-based term weighting framework DeepCT (Dai & Callan, 2019), and Contriever (Izacard et al., 2022) that is trained using unsupervised contrastive learning. We also compare our model with the state-of-the-art LLM-based retrieval model HyDE (Gao et al., 2023) which shares the exact same embedding spaces with Contriever but builds query vectors with LLMs.

**Methods with relevance judgment** Moreover, we also incorporate several systems that utilize fine-tuning on extensive query-document relevance data, such as MS-MARCO, as references (denoted as *w/ relevance judgment*). This group encompasses some commonly used fully-supervised retrieval methods, including DPR (Karpukhin et al., 2020), ANCE (Xiong et al., 2021), and the fine-tuned Contriever (Izacard et al., 2022) (denoted as Contriever[FT]).

### 5.3 IMPLEMENTATION DETAILS

As for the LLM part, we evaluate our proposed method on two options: closed-source models and open-source models. In the case of closed-source models, we employ the `gpt-3.5-turbo`, as it is popular and accessible to the general public[3]. As for the open-source models, our choice fell upon the Vicuna models (Chiang et al., 2023) derived from instruction tuning with LLaMa-1/2 (Touvron et al., 2023a;b). Specifically, we assessed the most promising 13B version of Vicuna from LLaMa-2, namely, `Vicuna-13B-v1.5`. Additionally, we evaluated the current best-performing 33B version of Vicuna derived from LLaMa-1, which is `Vicuna-33B-v1.3`. As for the RM part, we consider BM25 for retrieval since it is much faster. For each $q'$, we sample $h = 10$ knowledge examples via LLM. After hyper-parameter search on validation sets, we set $k$ as 15 for `gpt-3.5-turbo`, and 5 for `Vicuna-13B-v1.5` and `Vicuna-33B-v1.3`. We also set $M$ as 2 by default.[4] We use a temperature of 1 for LLM part in generation and a frequency penalty of zero. We also truncate each RM-retrieved

---

[3]We utilize the March 1, 2023 version of the `gpt-3.5-turbo` to avoid any interference caused by upgrading.

[4]For ArguAna data in BEIR benchmark, we set $M$ to 1 for `gpt-3.5-turbo` as it achieves the best performance.

Table 1: Experimental results on TREC Deep Learning 2019 (DL'19) and TREC Deep Learning 2020 (DL'20) datasets (%). The best results are marked in **bold** and the best performing w/ relevance judgment are marked with ¶. The improvement is statistically significant compared with the baselines w/o relevance judgment (t-test with $p$-value $< 0.05$)

| Methods | DL'19 | | | DL'20 | | |
|---|---|---|---|---|---|---|
| | MAP | nDCG@10 | R@1k | MAP | nDCG@10 | R@1k |
| *w/o relevance judgment* | | | | | | |
| BM25 (Robertson & Zaragoza, 2009) | 30.1 | 50.6 | 75.0 | 28.6 | 48.0 | 78.6 |
| Contriever (Izacard et al., 2022) | 24.0 | 44.5 | 74.6 | 24.0 | 42.1 | 75.4 |
| HyDE (Gao et al., 2023) | 41.8 | 61.3 | 88.0 | 38.2 | 57.9 | 84.4 |
| InteR (Vicuna-13B-v1.5 from LLaMa-2) | 43.5 | 66.4 | 84.7 | 39.4 | 57.1 | 85.2 |
| InteR (Vicuna-33B-v1.3 from LLaMa-1) | 45.8 | **68.9** | 85.6 | 45.1 | **64.0** | 87.9 |
| InteR (gpt-3.5-turbo) | **50.0** | 68.3 | **89.3** | **46.8** | 63.5 | **88.8** |
| *w/ relevance judgment* | | | | | | |
| DeepCT (Dai & Callan, 2019) | - | 55.1 | - | - | 55.6 | - |
| DPR (Karpukhin et al., 2020) | 36.5 | 62.2 | 76.9 | 41.8 | 65.3¶ | 81.4 |
| ANCE (Xiong et al., 2021) | 37.1 | 64.5¶ | 75.5 | 40.8 | 64.6 | 77.6 |
| Contriever^FT (Izacard et al., 2022) | 41.7¶ | 62.1 | 83.6¶ | 43.6¶ | 63.2 | 85.8¶ |

Table 2: Experimental results (nDCG@10) on low-resource tasks from BEIR (%). The best results are marked in **bold** and the best performing w/ relevance judgment are marked with ¶.

| Methods | SciFact | ArguAna | TREC-COVID | FiQA | DBPedia | TREC-NEWS |
|---|---|---|---|---|---|---|
| *w/o relevance judgment* | | | | | | |
| BM25 (Robertson & Zaragoza, 2009) | 67.9 | 39.7 | 59.5 | 23.6 | 31.8 | 39.5 |
| Contriever (Izacard et al., 2022) | 64.9 | 37.9 | 27.3 | 24.5 | 29.2 | 34.8 |
| HyDE (Gao et al., 2023) | 69.1 | **46.6** | 59.3 | **27.3** | 36.8 | 44.0 |
| InteR (Vicuna-13B-v1.5 from LLaMa-2) | 69.3 | 42.7 | **70.1** | 23.6 | 39.6 | 51.9 |
| InteR (Vicuna-33B-v1.3 from LLaMa-1) | 70.3 | 39.9 | 67.4 | 26.0 | 40.1 | 51.4 |
| InteR (gpt-3.5-turbo) | **71.7** | 40.9 | 69.7 | 26.0 | **42.1** | **52.8** |
| *w/ relevance judgment* | | | | | | |
| DPR (Karpukhin et al., 2020) | 31.8 | 17.5 | 33.2 | 29.5 | 26.3 | 16.1 |
| ANCE (Xiong et al., 2021) | 50.7 | 41.5 | 65.4¶ | 30.0 | 28.1 | 38.2 |
| Contriever^FT (Izacard et al., 2022) | 67.7¶ | 44.6¶ | 59.6 | 32.9¶ | 41.3¶ | 42.8¶ |

passage/document to 256 tokens and set the maximum number of tokens for each LLM-generated knowledge example to 256 for efficiency. *Source codes are uploaded for reproducibility.*

## 5.4 MAIN RESULTS

**Web Search** In Table 1, we show zero-shot retrieval results on TREC DL'19 and TREC DL'20 with baselines. We can find that InteR with selected LLMs can outperform state-of-the-art zero-shot baseline HyDE with significant improvement on most metrics. Specifically, InteR with gpt-3.5-turbo has an $> 8\%$ absolute MAP gain and $> 5\%$ absolute nDCG@10 gain on both web search benchmarks. Moreover, InteR is also superior to models with relevance judgment on most metrics, which verifies the generalization ability of InteR on large-scale retrieval. Note that our approach does not involve any training process and merely leverages off-the-shelf RMs and LLMs, which is simpler in practice but shown to be more effective.

**Low-Resource Retrieval** In Table 2, we also present the zero-shot retrieval results on six diverse low-resource retrieval tasks from BEIR benchmarks. Firstly, we find that InteR is especially competent on TREC-COVID and TREC-NEWS and even significantly outperforms baselines with relevance judgment. Secondly, InteR also brings considerable improvements to baselines on SciFact and DBPedia, which shows our performance advantages on fact-checking and entity retrieval. Finally, it can be observed that the performance of FiQA and ArguAna falls short when compared to the baseline models. This could potentially be attributed to the LLM's limited financial knowledge of FiQA and the RM's marginal qualification to effectively handle relatively longer queries for ArguAna (Thakur et al., 2021).

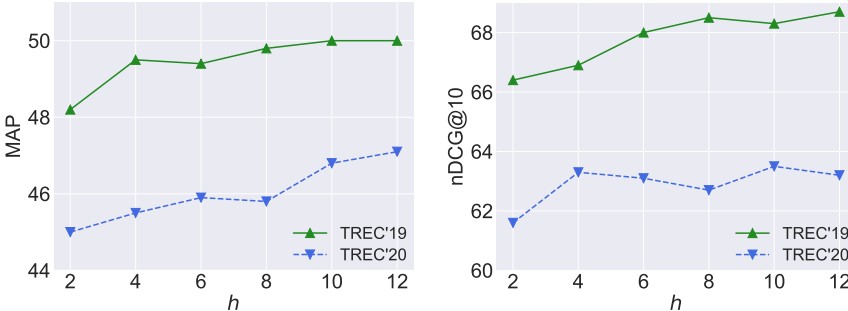

Figure 2: Performance of InteR with `gpt-3.5-turbo` across different size of knowledge collection ($h$) on TREC DL'19 and DL'20.

Table 3: Performance of InteR with `gpt-3.5-turbo` across different number of knowledge refinement iterations ($M$) on TREC DL'19 and DL'20. The default setting is marked with $*$ and the best results are marked in **bold**.

| Methods | DL'19 | | | DL'20 | | |
|---|---|---|---|---|---|---|
| | MAP | nDCG@10 | R@1k | MAP | nDCG@10 | R@1k |
| InteR ($M = 0$) | 30.1 | 50.6 | 75.0 | 28.6 | 48.0 | 78.6 |
| InteR ($M = 1$) | 45.8 | 65.3 | 89.3 | 42.6 | 61.0 | 88.7 |
| InteR ($M = 2$)* | **50.0** | **68.3** | **89.3** | **46.8** | **63.5** | **88.8** |
| InteR ($M = 3$) | 49.1 | 68.2 | 88.0 | 42.8 | 59.3 | 85.6 |

Table 4: Performance of InteR with `gpt-3.5-turbo` across different retrieval strategies for constructing $\bar{D}$ on Eq. 4 on TREC DL'19 and DL'20. The default setting is marked with $*$ and the best results are marked in **bold**.

| Methods | DL'19 | | | DL'20 | | |
|---|---|---|---|---|---|---|
| | MAP | nDCG@10 | R@1k | MAP | nDCG@10 | R@1k |
| InteR (Sparse) | 46.9 | 66.6 | **89.4** | 42.3 | 60.4 | 85.4 |
| InteR (Dense)* | **50.0** | **68.3** | 89.3 | **46.8** | **63.5** | **88.8** |
| InteR (Hybrid) | 48.3 | 67.6 | 89.1 | 45.1 | 62.5 | 85.2 |

## 5.5 DISCUSSIONS

**The impact of the size of knowledge collection** ($h$)   We conducted additional research to examine the impact of the size of knowledge collection (i.e., $h$) on the performance of InteR. Figure 2 illustrates the changes in MAP and nDCG@10 curves of InteR with `gpt-3.5-turbo` on TREC DL'19 and DL'20, with respect to varying numbers of knowledge examples. Our observations reveal a consistent pattern in both benchmarks: as the number of generated knowledge increases, the performance metrics demonstrate a gradual improvement until reaching 10 knowledge examples. Subsequently, the performance metrics stabilize, indicating that additional knowledge examples do not significantly enhance the results. This phenomenon could be attributed to the presence of redundant knowledge within the surplus examples generated by LLMs.

**The impact of the number of information refinement iterations** ($M$)   We also investigated the effect of different numbers of information refinement iterations ($M$) on the performance of InteR. The results of InteR with `gpt-3.5-turbo` presented in Table 3 indicate a notable enhancement in retrieval capacity as $M$ increases from 0 to 2, which verifies the effectiveness of multiple iterative information refinement between RMs and LLMs. However, if we further increase $M$, the performance may not improve, possibly due to a decrease in the diversity of retrieved documents from RMs. Here if we set $M$ to 0, InteR will degenerate into BM25.

**Dense retrieval v.s. sparse retrieval**   Furthermore, we delve into the impact of the retrieval strategy for constructing $\bar{D}$ on Eq. 4 on the performance of InteR. Table 4 shows the experimental results of

InteR with `gpt-3.5-turbo`, where we initiate the RM with an unsupervised sparse retrieval model (i.e., BM25) or an unsupervised dense retrieval model (i.e., Contriever). Additionally, we introduce a hybrid retrieval paradigm that combines half of the documents from the sparse retrieval model with the other half from the dense retrieval model. These components are then merged to generate the output of the hybrid retrieval model, namely InteR (Hybrid). We can observe that the dense model is the best strategy for InteR and even outperforms the hybrid counterpart, which is consistent with the observations from prior works (Karpukhin et al., 2020). Therefore, we use a dense retrieval model to construct $\bar{D}$ as default. It is noteworthy that we only use a dense retrieval model for constructing $\bar{D}$ during information refinement and leave the final retrieval model as a sparse BM25 for efficiency.

**Case Study on LLM Outputs**    To better illustrate how RMs enrich the outputs of LLMs, we selected an example from TREC DL'19 and compared those outputs of `gpt-3.5-turbo` after the first and second iterations. Due to the limited space, we only show the first LLM-generated knowledge (i.e., $s_1$). In the initial iteration ($M = 1$), the LLM demonstrated its capacity to generate paragraphs by leveraging its commonsense encoded within its parameters. After taking the search results of RM (documents about '*Thailand*') in the prompt, LLM outputs in the second iteration $M = 2$ became more informative. Specifically, the text highlighted in yellow elaborated on *Thailand's population and unemployment rate*, which was absent in the first iteration, and facilitated the next RM step.

---

**Case**: *query_id*: 1112341, *query*: "what is the daily life of thai people"

**LLM outputs of** $M = 1$**:** The daily life of Thai people is heavily influenced by their culture and traditions. The majority of the population practices Buddhism, and daily prayer and meditation are common. Family is also extremely important, and many Thai people live with extended family members. In terms of food, Thai cuisine is known for its spicy and savory flavors, and meals often revolve around rice and noodles. Street food is also a big part of daily life, with vendors selling snacks and meals on every corner. Work is an important aspect of daily life as well, and many Thai people work long hours in industries such as agriculture, tourism, and manufacturing. Despite the demands of work, there is a strong emphasis on taking breaks throughout the day to socialize and interact with others. Overall, the daily life of Thai people is a harmonious blend of religious practice, family values, flavorful cuisine, and hard work.

**LLM outputs of** $M = 2$**:** Thailand's daily life is influenced by its diverse geography and cultural influences from India, China, Cambodia, and the rest of Southeast Asia. The country's national religion, Theravada Buddhism, is central to modern Thai identity. Social occasions in Thailand often revolve around food, which is ordered and eaten in a communal style. Thai food presentation is among the most exquisite in the world, developed primarily in the palace to please the King of Siam. Thai people love to have fun together and often ask each other if they had a good time. The population of Thailand is approximately 67.5 million people, including ethnic groups such as Chinese, Malay, Lao, Burma, Cambodia, and Indians. Unemployment rate in Thailand is relatively low, standing at 0.84% according to Thailand's National Economic and Social Development Board (NESDB). Thai time is divided into five sections, with different names for each section of the day. The country's primary religion is Theravada Buddhism, and important holidays include Thai New Year, or Songkran. Overall, the daily life of Thai people is diverse, vibrant and deeply rooted in their rich cultural heritage.

---

# 6    CONCLUSION

In this work, we present **InteR**, a novel framework that harnesses the strengths of both large language models (LLMs) and retrieval models (RMs) to enhance information retrieval. By facilitating information refinement through synergy between LLMs and RMs, InteR achieves overall superior zero-shot retrieval performance compared to state-of-the-art methods, and even those using relevance judgment, on large-scale retrieval benchmarks involving web search and low-resource retrieval tasks. With its ability to leverage the benefits of both paradigms, InteR may present a potential direction for advancing information retrieval systems.

**Limitations**    While InteR demonstrates improved zero-shot retrieval performance, it should be noted that its effectiveness heavily relies on the quality of the used large language models (LLMs). If these underlying components contain biases, inaccuracies, or limitations in their training data, it could impact the reliability and generalizability of the retrieval results. In that case, one may need to design a more sophisticated method of information refinement, especially the prompt formulation part. We leave this exploration for future work.

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
