# OpenReview forum: "Synergistic Information Retrieval: Interplay between Search and Large Language Models"
_ICLR.cc/2024/Conference — Submitted to ICLR 2024_

### Official Review · Reviewer_nUtT · 2023-10-31

**Soundness:** 1 poor
**Presentation:** 3 good
**Contribution:** 1 poor
**Rating:** 3
**Confidence:** 5

**Summary:**

The paper leverages LLMs (templates) to iteratively generate passages more relevant to a given user query (sections 4.2 and 4.3). The INTER model then scores passages comparing the output of the LLM and documents using a dense model. Experiments on MS MARCO, and a set of 6 BEIR datasets, show that the model performs better than ANCE or Contriever-FT.

**Strengths:**

- A new method for query expansion relying on LLMs
- The method is zero-shot

**Weaknesses:**

First, the process depending on an LLM might not offer the same guarantees as trained retrieval models - as the LLM is a black box in this case.

Second, and more importantly, there is no comparison with sparse approaches (which have a good ZSL generalization capacity) or dense models (e.g. ColBERT-v2 or SPLADE-v2), for which the performance is much better than the proposed approach while being much lighter in terms of computation. The authors should also compare to BM25+monoT5/monoBERT since they generalize quite well on other datasets.

The argument of having a lightweight zero-shot system does not hold since these models perform at least on par with the proposed approach while being much lighter in terms of computation.

- ColBERTv2 http://arxiv.org/abs/2112.01488
- SPLADE https://doi.org/10.1145/3477495.3531857
- monoT5 https://aclanthology.org/2020.findings-emnlp.63/

**Questions:**

- How does the performance vary with different LLMs?
- What is the overall cost (computational) compared to other approaches?
- More experimental details on the effect of hyperparameters is needed given the very empirical nature of the approach

Other:

- DeepCT is not an unsupervised model

---

> ### Author Response · Authors · 2023-11-22
> **Response to Reviewer nUtT**
>
> We are extremely thankful for your valuable comments and suggestions.
>
> > **Question 1**: The process depending on an LLM might not offer the same guarantees as trained retrieval models, as the LLM is a black box.
> - Thanks for your great comment. On one hand, in this work we not only utilized the closed-source GPT but also employed the open-source Vicuna, hence it is not entirely a black box as we can get access to the model parameters. On the other hand, most present advanced LLMs are trained on large-scale web data, exhibiting a considerable degree of generalization and impressive performance on knowledge-intensive tasks. And some products even use LLMs for search (New Bing/Perplexity AI). Our experimental results also demonstrate that the LLM aids in the iterative information refinement for RM parts, and the overall framework achieves new state-of-the-art zero-shot retrieval performance.
>
>
> > **Question 2**: No comparison with sparse approaches
> - In our paper, we focus on zero-shot knowledge retrieval tasks and follow the experimental setup outlined in Gao et al. (2023) by establishing **zero-shot** sparse and dense retrieval baselines, as presented in Tables 1 and 2 for comparison. It is important to note that both ColBERT-v2 and SPLADE-v2 necessitate supervised training, thus requiring heavy query-document annotations. Nevertheless, SPLADE-max [1] achieves 68.4 and 85.1 on nDCG@10 and R@1k respectively on TREC DL 2019, which is worse than our method despite supervised training. Due to the limited time for rebuttal, we can not provide the performance of SPLADE-v2 now since neither the paper reports the performance on Trec DL nor releases their model. Of course, It is also possible to further enhance the performance by leveraging these supervised trained models. Thanks for your great comments!
>
>     - [1] Formal T, Lassance C, Piwowarski B, et al. SPLADE v2: Sparse lexical and expansion model for information retrieval[J]. arXiv preprint arXiv:2109.10086, 2021.
>
>
> > **Question 3**: How does the performance vary with different LLMs
> - The performance of different models depends on various factors such as training compute, model parameters, etc [2]. As depicted in Table-1, the open-source Vicuna-33B-v1.3 yields better results on DL'19 and DL'20 compared to the Vicuna-13B-v1.5 model (despite Vicuna-13B-v1.5 being from LLaMa-2 whereas Vicuna-33B-v1.3 is from LLaMa-1). This to some extent suggests that as the LM within InteR grows larger, the effectiveness of zero-shot retrieval improves.
>
>     - [2] Jared Kaplan et al., 2020. Scaling laws for neural language models. arXiv preprint arXiv:2001.08361
>
> > **Question 4**: About the overall computational cost
> - Thank you for your constructive suggestions! As the RM part utilizes BM25 which is much faster, the primary bottleneck lies within the LM. For closed-source ChatGPT models, the latency cannot be accurately ascertained due to the API nature. Hence, we additionally included the latency result of open-source models Vicuna-13B-v1.5 on 1 GPU of A100-40GB for each iteration using vLLM [3] in our revised manuscript. Then we observed that it can handle 20.16 requests per minute, which can be further optimized with more advanced LM inference tools.
>
>     - [3] https://github.com/vllm-project/vllm
>
> > **Question 5**: More experimental details on hyperparameters”
> - Thank you for your suggestion. We tuned the hyperparameters on validation sets of TREC DL data. We have included these details in our revised manuscript.
>
> > **Question 6**: DeepCT
> - Thank you for your reminder, we have corrected this mistake in our revision.

---

### Official Review · Reviewer_8fe1 · 2023-11-03

**Soundness:** 3 good
**Presentation:** 3 good
**Contribution:** 3 good
**Rating:** 6
**Confidence:** 4

**Summary:**

The paper presents InterR, an information retrieval method that refines the query iteratively using an LLM and a retriever model. Both modules update the query by adding new relevant information to it. When using Vicuna/gpt-3.5 in combination with BM25, the method exhibits competence against zero-shot models as well as those based on relevance judgements on web search and low-resource scenarios. Ablation and case studies are included to analyze the method.

**Strengths:**

- The method is simple yet very effective. The results are strong.
- The idea is not super novel but original to the best of my knowledge.
- Reproducibility: The authors promise to release the code.
- Clarity: The paper is overall well-written.

**Weaknesses:**

- The method achieves strong accuracy, but its latency must have been severely compromised as it involves multiple rounds of LLM and RM querying. This makes this method unideal for most retrieval applications where latency is prioritized. Would be great to include the latency results so that readers can better understand the tradeoff.
- One part that is missing is an analysis of the side- or negative- impacts of the method. The iterative expansion will likely sometimes introduce information with negative effects. e.g. irrelevant info, contradictory logics, hallucinated content, lexically similar but semantically irrelevant texts, etc. How often does this happen? How does this hurt performance? Any potential mitigation strategy?

**Questions:**

- Are the authors from Microsoft? If so, there is a potential **RISK OF IDENTITY LEAK** as they use the Bing logo in Figure 1 but the experiment setting is totally irrelevant to Bing. I am just pointing out the risk here and will let the meta-reviewer decide whether this is an issue or not.
- On what dataset do you tune the hyperparameters on? If it is the test set, then it largely weakens the reliability of the conclusions.
- Another area of related work to cite and discuss is query rewriting in conversational QA (e.g. QReCC), where a NLG model is used to refine the query to augment retrieval.

---

> ### Author Response · Authors · 2023-11-22
> **Response to Reviewer 8fe1**
>
> We are extremely thankful for your valuable comments.
>
> > **Question 1**: About the latency results
> - Thank you for your constructive suggestions! As the RM part utilizes BM25 which is much faster, the primary bottleneck lies within the LM. For closed-source ChatGPT models, the latency cannot be accurately ascertained due to the API nature. Hence, we additionally included the latency result of open-source models Vicuna-13B-v1.5 on 1 GPU of A100-40GB for each iteration using vLLM [1] in our revised manuscript. Then we observed that it can handle 20.16 requests per minute, which can be further optimized with more advanced LM inference tools.
>
>     - [1] https://github.com/vllm-project/vllm
>
>
> > **Question 2**: The iterative expansion will likely sometimes introduce information with negative effects.
> - Thanks for your insightful comments and suggestions!  We acknowledge that iterative information refinement can sometimes lead to negative effects such as the introduction of irrelevant information and hallucinated content. As outlined in the Limitation Section, we have also identified that enhancing the quality control in prompt formulation and query expansion is crucial for further improving the model. Since our study aimed to validate the effectiveness and universality of InteR framework, we did not excessively design this part and retained only the most fundamental components. We are glad to investigate this to future improve our method in the future. Thank you again for your great suggestions.
>
>
> > **Question 3**: Logo in Figure 1
> - Thanks for your kind reminder and sorry for this misunderstanding. We randomly selected a commonly used search engine for daily purposes for the logo. We have rectified this in the revised version. Thank you once again for the kind reminder!
>
> > **Question 4**: Hyperparameters tuning method & missed references
> - Thank you for the kind reminder! We tuned the hyperparameters on validation sets of TREC DL data. We have also included the missed references in our revision.

---

### Official Review · Reviewer_zzZ7 · 2023-11-03

**Soundness:** 3 good
**Presentation:** 3 good
**Contribution:** 2 fair
**Rating:** 5
**Confidence:** 5

**Summary:**

Information retrieval (IR) is a critical technique to locate relevant data in vast collections. Modern search engines (SEs) are based on neural representation learning and large language models (LLMs), which provide contextual understanding and the potential to save time by directly answering queries. However, LLMs have limitations, such as generating outdated or incorrect information, whereas search engines can swiftly sift through vast updated data.  This paper intends to bridge the capabilities of retrieval models (RMs) and LLMs, enhances IR by expanding queries using LLM-generated knowledge and enriching LLM prompts using RM-retrieved documents.

**Strengths:**

1. Introduction of InteR, a framework that combines the strengths of both RMs and LLMs, addressing their individual limitations.

2.  LLMs can understand user-issued queries' context, generating specific answers rather than just presenting a list of relevant documents, RMs can provide fresh information.

3. The experimental results showcasing that InteR can perform zero-shot retrieval more effectively than other state-of-the-art methods.

**Weaknesses:**

1. The idea is not new, as You.com and Bing.com are doing this, and Google Bard undoubtedly has developed InTeR-like capabilities.

2. RMs require well-crafted keywords to deliver accurate results, which might not be user-friendly.  Query formulation is a problem
to be addressed.

3. The extensive reliance on vast training datasets like NQ and MS-MARCO for dense retrieval models, which can be time-consuming and may have commercial use limitations.

**Questions:**

1. How scalable is the InteR framework when applied to different types of data beyond the current experimental setup?

2. Can there be further optimizations in the synergy between RMs and LLMs to address real-time search requirements?

---

> ### Author Response · Authors · 2023-11-22
> **Response to Reviewer zzZ7**
>
> > **Question 1**: Comparison with You.com and Bing.com & The idea is not new
> - Thank you for your valuable comment. To the best of our knowledge, we are not acquainted with the technical details of these two products, as neither product discloses its technical details nor releases technical reports. Combining retrieval models and LLM is a popular direction now, while most existing works only pursue single-step/mono-direction enhancement like “Enhance Retrieval Through LMs” or “Enhance LMs Through Retrieval”. In our work, we are among the earliest to propose iterative information refinement through synergy between retrieval models and LLMs, resulting in more accurate retrieval than state-of-the-art approaches and even outperforming baselines that leverage relevance judgment for supervised learning. We humbly believe that our paper has made a unique contribution and may offer insights for future research in this field.
>
> > **Question 2**: Query formulation is a problem to be addressed
> - Yes, because the query requires being well-crafted, the retrieval performance is not satisfactory. Therefore, we employ LLM for iterative information refinement of RM queries, aiding in better comprehension of the query. Query formulation, specifically designed in our proposed framework, is indeed adopted to improve retrieval results.
>
> > **Question 3**: Extensive reliance on vast training datasets can be time-consuming and may have commercial use limitations
> - Thanks for your valuable comments. Our paper focuses on zero-shot retrieval, hence, we do not rely on any training datasets such as NQ and MS-MARCO. As stated in Section 5.3, "As for the RM part, we consider BM25 for retrieval since it is much faster." The proposed models involved in our primary experiment are all based on BM25 and do not utilize the vast training datasets. Even the Contriever mentioned in Section 5.5 is used for zero-shot retrieval. Section 5.3 includes our training details, and we hope the above clarification can resolve your concerns.
>
>
> > **Question 4**: How scalable is the InteR framework when applied to different types of data.
> - Thank you for your valuable suggestions! Indeed, our approach can be easily adapted to other knowledge-intensive tasks such as QA and fact-checking. We leave these explorations to future work.
>
>
> > **Question 5**: How to optimize our framework for real-time search
> - Thank you for your great question! Our approach sacrificed some efficiency to enhance effectiveness, which is a trade-off. To address real-time search, further optimization can be implemented: Firstly, facilitating real-time search can be achieved by reducing the number of knowledge refinement iterations (M). As shown in Table 3, even with M=1, the retrieval performance remains satisfactory. Since the time required for retrieval grows linearly with M, and the primary bottleneck lies within the LM (whereas the RM utilizing BM25 is much faster than LM), we can further optimize by enabling the LM to decide in a single pass whether to terminate further iterations or generate an answer, thereby concurrently managing both sub-tasks. Lastly, certain products like vLLM[1] have already explored various efficient techniques for leveraging LMs in inference,  and some search engines (like Bing and Google) have tried to incorporate LLMs to improve search quality. All the above efforts can facilitate real-time search. Thanks again for your great question!
>
>     [1] https://github.com/vllm-project/vllm

---

> > ### Comment · Reviewer_zzZ7 · 2023-11-22
> >
> > I acknowledge reading the authors' rebuttals, and will maintain my rating.

---

> > > ### Author Response · Authors · 2023-11-22
> > >
> > > Thank you for reviewing our rebuttals and for your continued consideration of our submission. While we understand and respect your decision to maintain the current rating, we remain open to any additional feedback or suggestions that could further improve our work.

---

### Official Review · Reviewer_gmGa · 2023-11-05

[review text omitted: it was posted to a different submission]

---

> ### Author Response · Authors · 2023-11-22
> **Response to Reviewer gmGa**
>
> We sincerely appreciate your thoughtful review. However, upon careful consideration, we noted a potential misalignment between the review and our manuscript. Our paper does not explicitly emphasize 'proposing a Bayesian tool' as mentioned in your review. We kindly request that you check whether the review aligns with this manuscript. Thank you once again for your time and valuable feedback.

---

> > ### Comment · Reviewer_gmGa · 2023-11-22
> > **Misalignment between the review and the manuscript**
> >
> > Accidentally the review appearing here is not for this paper but for an other paper assigned to me. The review fro this paper "Synergistic Information Retrieval: Interplay between Search and Large Language Models" is also linked to an other paper.

---

### Meta-Review · Area_Chair_hr7N · 2023-12-29

**Metareview:**

The authors introduce InteR, a method that iteratively reformulates the user query for input to the IR system using a LLM (e.g., GPT, Vicuna) and uses the IR system to extract passages to include in later rounds of answer refinement -- with the goal of improving IR (i.e., not RAG-style generation goals). InteR is empirically validated in zero-shot and fine-tuned variants of sparse (e.g., BM25, DeepCT) and dense (e.g., DPR, Contriever, HyDE) retrieval models on widely-used web search benchmarks (e.g., TREC DL) and low-resource retrieval settings (SciFact, ArguAna, TREC-Covid, FiQA, DBPedia, TREC-News) -- showing consistent improvements over recent work (with HyDE being the most relevant competing method). Additionally, sensitivity analyses are performed regarding the size of the knowledge collection, number of refinement iterations, and dense vs. sparse retrieval in the InteR loop. Finally, a case study is performed to show effects on the actual LLM response generation task.

Consensus strengths identified by reviewers regarding this submission include:
- It makes sense to use LLMs for query expansion/augmentation and the proposed method is relatively straightforward and shown to perform well. The IR/SE component can focus on precision and the LLM can add some recall-inducing content.
- The experimental results demonstrate that InteR performs well relative to strong recent baselines.
- The paper is well-written in the sense that the proposed method is clearly described and easily understood.

Conversely, consensus limitations included:
- While there isn't a technical report or publication to directly refer to, the reviewers point out that many people are almost certainly doing this in commercial settings (which I can vouch for). Therefore, the contribution is an academic/public version of this done in a more rigorous fashion. Thus, while technically novel, it isn't a notably creative formulation.
- The number of rounds used (after hyper-parameter selection) is M=2, which should be examined further. Is the cause of poorer performance at M=3 irrelevant information being introduced? Can a different prompt work better in later rounds? With M=2, this seems very close to a LLM-based query expansion and not a multi-round solution. (and thus seems preliminary)
- There should be some discussion regarding latency in practical settings. While query expansion is used in practice and this was discussed in the rebuttal (and showing BM25 is sufficient with InteR is useful), it is a valid concern that should be discussed further.
- One reviewer mentioned zero-shot sparse approaches as a comparison, which I believe was adequately rebutted, but should be addressed at least through discussion in a final manuscript.

In my own reading, my additional concerns include (and I also concur with the reviewer concerns):
- In Section 2, the authors should clearly state how InteR contrasts with the related work. As it is written, reading the HyDE paper is almost mandatory IMO.
- There is a strong relationship with traditional query expansion methods (e.g., use external resources or retrieved documents) that isn't sufficiently discussed. Furthermore, these maybe should even be included as baselines as InteR is the only method that uses multiple rounds (I believe) and thus has an unfair advantage in a way.
- The evaluation is performed on IR tasks, but the case study is more of a RAG case. It would seem that the case study should be looking at what is retrieved better using the method.

Overall, I think this is a useful contribution in establishing a published baseline method for performing IR query expansion with a SotA LLM. However, I also believe that without more 'deep dive' experiments and associated discussion, the current empirical results are promising, but preliminary.

**Justification For Why Not Higher Score:**

The conceptual novelty is limited even if there isn't a published report to compare with; thus, I think more variants are necessary to make for a more complete baseline. However, more importantly, additional experiments are needed to understand the dynamics of the performance. Is this basically 'just' using LLMs for a round of query expansion or is it really an iterative method as the paper claims in the abstract/title/etc.

**Justification For Why Not Lower Score:**

N/A

---

### Decision · Program_Chairs · 2024-01-16

Reject